# Ultrasound Assisted Synthesis and In Silico Modelling of 1,2,4-Triazole Coupled Acetamide Derivatives of 2-(4-Isobutylphenyl)propanoic acid as Potential Anticancer Agents

**DOI:** 10.3390/molecules27227984

**Published:** 2022-11-17

**Authors:** Sadaf Mahmood, Samreen Gul Khan, Azhar Rasul, Jørn Bolstad Christensen, Mohammed A. S. Abourehab

**Affiliations:** 1Drug Design and Medicinal Chemistry Laboratory, Department of Chemistry, Government College University, Faisalabad 38000, Pakistan; 2Department of Zoology, Faculty of Life Sciences, Government College University Faisalabad, Faisalabad 38000, Pakistan; 3Department of Chemistry, Faculty of Science, University of Copenhagen, Frederiksberg C, 1870 Copenhagen, Denmark; 4Department of Pharmaceutics College of Pharmacy, Umm Al-Qura University, Makkah 21955, Saudi Arabia

**Keywords:** 2-(4-isobutylphenyl)propanoic acid, acetamides, triazole, in silico modelling, liver cancer

## Abstract

The development of an economical method for the synthesis of biologically active compounds was the major goal of this research. In the present study, we have reported the ultrasound-radiation-assisted synthesis of a series of novel *N-*substituted 1,2,4-triazole-2-thiol derivatives. The target compounds **6a–f** were efficiently synthesized in significant yields (75–89%) by coupling 1,2,4-triazole of 2-(4-isobutylphenyl) propanoic acid **1** with different electrophiles using ultrasound radiation under different temperatures. The sonication process accelerated the rate of the reaction as well as yielded all derivatives compared to conventional methods. All derivatives were confirmed by spectroscopic (FTIR, ^1^HNMR, ^13^CNMR, HRMS) and physiochemical methods. All derivatives were further screened for their anticancer effects against the HepG2 cell line. Compound **6d** containing two electron-donating methyl moieties demonstrated the most significant anti-proliferative activity with an IC_50_ value of 13.004 µg/mL, while compound **6e** showed the lowest potency with an IC_50_ value of 28.399 µg/mL. The order of anticancer activity was found to be: **6d** > **6b** > **6f** > **6a** > **6c** > **6e**, respectively. The in silico modelling of all derivatives was performed against five different protein targets and the results were consistent with the biological activities. Ligand **6d** showed the best binding affinity with the Protein Kinase B (Akt) pocket with the lowest ∆G value of −176.152 kcal/mol. Compound **6d** has been identified as a promising candidate for treatment of liver cancer.

## 1. Introduction

Cancer and other infectious diseases are significant health concerns worldwide [1]. Cancer mortality rates will more than double in the coming years, according to the World Health Organization [2]. After receiving chemotherapy, cancer patients became vulnerable to microbial infections due to the emergence of multidrug resistance, a weak immune system and the development of resistant strains of pathogenic microorganisms [2,3,4]. Therefore, the development of multi-target drugs acting as both anticancer and antimicrobial agents is a major focus of researchers globally. There is a need for intensive efforts to discover and develop new anticancer drugs with high safety profiles [5].

Nitrogen-based heterocyclic compounds have played an important role in the field of pharmaceuticals [6]. Approximately 75% of FDA-approved drugs are nitrogen-based heterocyclic compounds [7]. These heterocycles are reported to have great importance in medicinal chemistry [8]. According to the pharmacodynamical principle of superposition, 1,2,4-triazole, a versatile scaffold, is reported to have potential pharmacological activities, such as anticancer [9], antifungal [10], anti-urease [11], anti-inflammatory [12], antioxidant [13] as well as anticonvulsant [14] properties. Acetamide is reported to be a significant and important pharmacophore of anti-cancer drugs [15]. Cefatrizine and Tazobactam are triazole-based antimicrobial and anticancer drugs, respectively (Figure 1).

The scientific community is facing major challenges in the development of environmentally sustainable and economical techniques for the synthesis of novel pharmacological agents. Sonochemistry is a technique for accelerating the rate of chemical reactions by using ultrasound radiation in the range of 20 KHz to 100 KHz in a sonicator [16,17,18]. This technique offers a fast, efficient and economic procedure for the synthesis of bioactive molecules in good yields within a short amount of time [19]. In contrast to conventional heating techniques or systems involving catalysts, ultrasound radiation accelerates the rate of various chemical reactions through the phenomenon of cavitation [20,21]. The bubbles produced during the cavitation phenomenon collapse with the sound waves, generating a tremendous amount of energy and transforming kinetic energy into heating energy [22].

Molecular hybridization is a simple and effective tool to covalently combine multiple drug pharmacophores. On this basis, we have designed a hybrid of triazole and acetamide pharmacophores together in one molecule to prevent the progression of cancer. We selected 2-(4-isobutylphenyl)propanoic acid core for our chemical derivatization. Our designed molecular framework contains hydrophobic aryl rings on each side of the triazole ring to increase its bioavailability and cell penetration.

Previously, we have synthesized different pharmacologically active oxadiazole-based polyvalent derivatives containing acetamide functionalities [23,24,25]. So, in continuation of our previous work on heterocyclic compounds, we have cyclized the –COOH group of 2-(4-isobutylphenyl)propanoic acid into a five-membered heterocycle, e.g., triazole nucleus.

## 2. Results

### 2.1. Chemistry

The present work is a synthetic approach for the ultrasound-assisted synthesis of new *N-*substituted 5-aryl-1,2,4-triazole-3-acetamide derivatives **6a–f**. Ultrasound-assisted reactions are economical in terms of having good yields and shorter periods of time. The synthetic route of synthesized compounds **6a–6f** is depicted in Figure 1. These reactions are based upon the chemical modification of 2-(4-isobutylphenyl)propanoic acid by using ultrasound radiation as the energy source required for the completion of the reaction. Triazole was derivatized at SH by replacing its *H*-group with different electrophiles, resulting in the final compounds **6a–6f** in good yields.

Compound **2** was obtained by refluxing the 2-(4-isobutylphenyl)propanoic (**1**) with absolute methanol for six hours using the Fischer esterification method [26]. The ester of 2-(4-isobutylphenyl)propanoic acid was gently refluxed with 80% hydrazine hydrate [27] in methanol to obtain 2-(4-isobutylphenyl)propane hydrazide (**3**). In the third step, compound **3** was cyclized into 5-(1-(4-isobutylphenyl) ethyl)-1,2,4-triazole-2-thiol (**4**) by gently refluxing compound **3** in methanol with methyl isothiocyanate in 10% NaOH. After this was completed, the reaction mixture was acidified by concentrated HCl, and product **4** was isolated by filtration.

Triazole derivatives **6a–6f** were achieved by coupling compound **4** with different substituted aralkyl/alkyl/aryl 2-bromoacetamides using ultrasound radiations for 39–80 min at 45–55°C. The conventional approach of synthesizing triazole derivatives was also used, which resulted in all the derivatives in moderate yields over a longer time period (10–36 h). The ultrasound-assisted strategy was more efficient in terms of the maximum yield (65–80%) and reduced the time, as compared to conventional methods. The structures of triazole-based acetamides were confirmed by spectroscopic (FTIR, ^1^HNMR, ^13^CNMR) and physiochemical methods.

The structures of molecules were verified by different spectroscopic techniques (Appendix A) and their purity was confirmed by the molecular ion peaks in the HRMS spectra (Appendix A). In the IR spectrum, the triazole-coupled acetamide formation was confirmed by the appearance of absorption bands of NH stretching around (3282–3235 cm^−1^), aromatic C-H vibrations (3097–3032 cm^−1^), C = O in the acetamide peak (1700–1635 cm^−1^), N = N stretching (1677–1650 cm^−1^), C-N stretching of the acetamide (1310–1230 cm^−1^), and C-N stretching of the triazole (1085–1075 cm^−1^). In the ^1^HNMR spectrum of **6a–6f**, the protons of the aromatic ring system were observed in between 7.75 and 6.93 ppm. The aliphatic protons of propanoic acid were the most shielded and were observed to be around 0.86–0.84 ppm. Key evidence of the formation of an acetamide moiety was the appearance of the CH_2_ peak at 4.04 ppm and the NH peak around 10.27–9.64 ppm. ^13^CNMR spectroscopy also supported the formation of acetamide by the emergence of peaks around 166.12–164.18 ppm that correspond to the C = O group of acetamide, and peaks at 158.15 ppm that correspond to the carbon nucleus of the triazole ring. The synthesis of triazole was further confirmed by the appearance of N-CH_3_ at 3.28 ppm in ^1^HNMR and at 29.58 ppm in ^13^CNMR. The characteristic signals at 22 ppm corresponded to two -CH_3_ carbon nuclei of propanoic acid. The segregation of signals between 44.18 and 20.19 ppm corresponded to CH_2_ carbon of propanoic acid. All these peaks confirmed the synthesis of triazole-based molecules.

### 2.2. Anti-Proliferative Potential

The anticancer potential of all *N*-substituted 5-aryl-1,2,4-triazole-3-acetamide derivatives **6a–6f** was checked against the liver cancer HepG2 using an MTT assay [28]. All compounds showed moderate to excellent anticancer activity, which varied according to the substitution on the aryl rings, as shown in (Table 1). The MTT assay after 48 h showed the rate of the proliferation of HepG2 cells after exposure to different concentrations (3.25 µg/mL to 100 µg/mL) of the samples **6a–6f** (Table 2). The positive control shows the 100% viability of the HepG2 cells without drug exposure. A comparison of all these samples revealed the most significant cytotoxicity was exhibited by **6d** with a minimum IC_50_ value of 13.004 µg/mL (Table 1). While compound **6e** showed the lowest potency and highest IC_50_ value of 28.399, compound **6e** containing the electron-withdrawing bromo group was the least active. The order of activity was found to be: 2,3-Dimethyl phenyl ≥ 2,4-Dimethyl phenyl ≥ 3-Methyl phenyl ≥ 4-Methyl phenyl ≥ 2-Bromo phenyl ≥ 4-Bromo phenyl, respectively (Figure 2).

The MTT assay graph after 48 hours showed the rate of the proliferation of the HepG2 cells after exposure to different concentrations (3.25 µg/mL to 200 µg/mL). Compounds **6d** and **6b** showed the most significant cell viability at 12.5 µg/mL (Table 2 and Figure 2). The MTT assay graph after 48 hours showed the percentage of the age inhibition of the proliferation of the HepG2 cells after exposure to different concentrations (3.25 µg/mL to 200 µg/mL). Compounds **6d** and **6b** showed most significant percentage of age inhibition at 12.5 µg/mL.

### 2.3. Structure–Activity Relationship (SAR) Study

The anticancer activity of all compounds **6a–6f** was checked against the liver cancer HepG2 cell lines by using an MTT assay. Amongst all the compounds, **6d** was most potent, containing two electron-donating methyl substituents at the ortho and meta-position of the aryl ring. Meanwhile, compound **6e** with an electron-withdrawing bromo group at the para position showed the lowest potency and highest IC_50_ value, 28.399.

The order of decreasing anti-proliferative activity was as follows: **6d** > **6b** > **6f** > **6a** > **6c** > **6e**, respectively. Based on the findings from the SAR of *N*-substituted 5-aryl-1,2,4-triazole-3-acetamide derivatives **6a–6f**, it is concluded that the presence of the electron-donating -CH_3_ motif at the ortho- and meta-position of the aryl ring enhanced the anti-proliferative activity of the compound.

### 2.4. Molecular Docking

The in silico modelling of **6a–6f** was carried out to examine the potential mode of action of the compounds under study that may be liable for their anti-cancer activities. Five important targets, Signal transducer and activator of transcription 3 (STAT3), Phosphatidylinositol 3-kinase alpha (PI3Kalpha), c-Kit Tyrosine Kinase (c-Kit), Protein kinase B (Akt) and Human Aurora B kinase, were selected as important proteins in cancer treatment. Signal transducer and activator of transcription 3 (STAT3) is a secret transcription factor that is activated in response to different cytokines, growth factors and oncogene signals. Therefore, it is considered as an attractive cancer therapeutic target. Phosphatidylinositol 3-kinases alpha are members of the intracellular lipid kinases that regulate cell metabolism, survival, growth and proliferation. They have been associated with multiple human cancers. Protein kinase B (Akt) plays a considerable role in signaling within cells, by promoting both cell proliferation and survival, and in the case of cancer, this path is irregular. Human Aurora kinase B has an important role in the cell cycle and is overexpressed in tumor cells. The potential effects of the molecules on the targets were investigated using the web page: http://www.swisstargetprediction.ch/ [29,30,31,32,33,34]. Protein kinase B (Akt) plays a considerable role in signaling within cells as well as promoting both cell proliferation and survival, and in the case of cancer, this path is irregular. The results suggested and supported the effectiveness of the molecules against the kinase targets.

The selected targets, their codes in the protein data bank, the grid box coordinates and the molecular docking scores are presented in Table 3. The docking results show that the molecules have the potential to affect different targets.

All compounds showed better activity against STAT3, P13 Kalpha and Human A. c-Kit Tyrosine Kinase and Protein Kinase B (Akt) were predicted to have a strong interaction potential against the cancer therapeutic targets with the best binding poses and a low binding energy. These results suggested and supported the effectiveness of the molecules against the kinase targets.

The docking results showed that all the molecules have the potential to affect the Protein Kinase B target. Protein Kinase B (Akt) is predicted to have a strong interaction potential against cancer therapeutic targets. The binding affinities, categories, types of interactions and interacting residues of the docked complexes were assessed, as presented in Table 3. The hydrogen bonding interactions and hydrophobic contacts of every ligand were evaluating inside the binding site of the receptor protein. The ligands conformations which demonstrated the maximal biological activities are explained in Table 4 and Figure 3 with their favorable interactions in the binding sites.

As demonstrated in Table 4, compounds **6d** and **6b** are bound to the catalytic site of Protein Kinase B with the best binding poses and lowest binding energy, −176.152 and −169.631, respectively. The molecule **6d** bound to Protein Kinase B by forming H-bond interactions with Lys181. It interacted with the residues Leu 296 and His 196 through alkyl interactions with Ile188, Lys160, Gly161, Phe163, Gly164, Lys165, Thr162, Gly159, Glu236, Glu279, Arg(c)6, Asp293, and Asn280 residues involving the van der Waals interactions between compound **6d** and Protein Kinase B (Figure 4).

The molecule **6b** bound to Protein Kinase B by forming H-bond interactions with Ser 9 and Lys 277, and Sulfur-x interactions with Thr 7. It interacted with the residues Leu183, Lys181 and Val166 through alkyl interactions (Figure 5).

The docked conformations of ligands **6a–f** and the reference ligand STI were examined to obtain a qualitative estimation and to identify the molecular foundation of the analyzed biological activities (Figure 6).

As demonstrated in Table 5, compounds **6d** and **6b** are bound to the catalytic site of Protein Kinase B with the best binding poses and lowest binding energy, −180.052 and −170.726, respectively. Compound **6d**’s key interactions with Protein c-Kit Tyrosine Kinase involved the H-bond (Cys673), Pi-doner H-bond (Thr670), Pi-Sigma bond (Leu595, Leu644), Alkyl (Leu644, Val654, Cys809, Leu799, Lys623, Phe811, Val668, Val603, Ala621), and van der Walls (Glu 671, Ala621, Glu640, val643, leu647, Ile808, Leu783, Ile653, His790, Asp810, Thr670, Val603, Asp677, Gly676) interactions (Figure 7).

The molecule **6b** bound to Protein c-Kit Tyrosine Kinase by forming H-bond interactions with Glu640 and Thr670. Other key interactions involved hydrophobic alkyl interactions with the residues Val603, Ala621, Val643, Val654, Leu644 and Val668 (Figure 8).

## 3. Materials and Methods

### 3.1. General

Melting points of all derivatives were checked using open capillary tube methods and an electrical melting point (Stuart SMP10 melting point apparatus). As a starting material, 2-(4-isobutylphenyl)propanoic acid was used. Ultrasonic irradiation synthesis was performed in an ultrasonic cleaner bath (Model 1510, 1.9 L, 115 v) at frequency of 47 kHz. Infrared spectra were obtained using potassium bromide discs on an FT-IR spectrophotometer (BRUKER) with a wavelength range of 4000–400 cm^−1^ and were expressed in wave numbers (cm^−1^) at GC University’s Hi Tech Lab. ^1^HNMR and ^13^CNMR spectra were carried out on Advance Bruker using 500 MHz spectrophotometer (^1^HNMR) and 75 MHz to 126 MHz (^13^CNMR) spectrophotometer at Department of Chemistry, University of Copenhagen, Denmark. The chemical shift was expressed in δ ppm. Reaction progress and completion was monitored by thin-layer chromatography.

### 3.2. General Procedure for the Synthesis of Synthesized N-substituted 5-aryl-1,2,4-triazole-3-acetamide derivatives ***6a–6f***

#### 3.2.1. Synthesis of methyl 2-(4-isobutylphenyl)propanoate (**2**)

Compound **2** was synthesized by the method reported in [23]. Compound **2** was obtained as pale yellow oily liquid with the following properties: yield 90%; b.p. 263–265 °C; IR (KBr) cm^−1^: 1736.34, 1203.37, 1162.81; and ^1^HNMR (400 MHz, CDCl_3_) δ 7.15 (d, 2H, *J* = 8.0 Hz, H-3 and H-5), 7.05 (d, 2H, *J* = 8.0 Hz, H-2 and H-6), 3.67 (q, 1H, H-7), 3.60(s, 3H, OCH_3_), 2.40 (d, 2H, *J* = 8.0 Hz, H-9), 1.82 (m,1H, H-9), 1.44 (d, 3H, *J* = 8.0 Hz, H-8), 0.85 (d, 6H, *J* = 8.0 Hz, H-11 and H-12). ^13^C NMR (101 MHz, CDCl_3_) δ 175.22, 140.49, 137.63, 129.04, 127.31, 52.09, 44.99, 30.25, 22.21, 18.50. HRMS (ESI+): m/z calculated for [(C_14_H_20_O_2_)+H]^+^: 220.1463; found: 220.1460. Element analysis calculated: C, 76.33; H, 9.15; observed: C, 76.30; H, 9.16 %.

#### 3.2.2. Synthesis of 2-(4-isobutylphenyl)propane hydrazide (**3**)

Compound **3** was synthesized by the method reported in [23]. 2-(4-isobutylphenyl) propane hydrazide was separated as a white crystalline solid. Yield: (88%); m.p. 77–78 °C; IR (KBr) cm^−1^: 3272.75, 2963.11, 1640.11, 1604.85, 1466.29, 1366.60, 906.66, 686.81. ^1^H NMR (400 MHz, CDCl_3_) δ 9.50 (s, 1H, NH), 7.13 (d, 2H, *J* = 8.0 Hz, H-3 and H-5), 7.06 (d, 2H, *J* = 8.0 Hz, H-2 and H-6), 3.48 (d, 2H, NH_2_), 3.46 (q, 1H, H-7), 2.40 (d, 2H, *J* = 8.0 Hz, H-9), 1.81 (m,1H, H-9), 1.48 (d, 3H, *J* = 8.0 Hz, H-8), 0.84 (d, 6H, *J* = 8.0 Hz, H-11 and H-12). ^13^C NMR (101 MHz, CDCl_3_) δ 175.24, 140.49, 137.59, 129.57, 127.30, 44.96, 30.25, 22.19, 18.23. HRMS (ESI+): m/z calculated for [(C_13_H_20_N_2_O)+H]^+^. 220.1576; found: 220.1574. Element analysis calculated: C, 70.87; H, 9.15; N, 12.72; observed: C, 70.85; H, 9.16; N, 12.72 %.

#### 3.2.3. Synthesis of 5-(1-(4-isobutylphenyl) ethyl)-1,2,4-triazole-2-thiol (**4**)

In an oven-dried round bottom flask, 2-(4-isobutylphenyl) propane hydrazide (0.02 mol) was mixed with equimolar amount of methyl isothiocyanate in 10% KOH solution. The mixture was refluxed at 95 °C for 10–11 h. Reaction progress as well as completion was continuously monitored by TLC. After completion, chilled water was added in flask to obtain precipitates of product. Precipitates were filtered and washed with water. The product was also recrystallized using absolute ethanol. 5-(1-(4-isobutylphenyl)ethyl)-1,2,4- triazole-2-thiol was separated as off-white crystalline solid. ^1^HNMR (400 MHz, CDCl_3_) δ 11.77 (s, 1H, SH), 7.05 (d, 2H, *J* = 8.0 Hz, H-3 and H-5), 6.99 (d, 2H, *J* = 8.0 Hz, H-2 and H-6), 3.96 (q, 1H, H-7), 3.18 (s, 3H, N-CH_3_), 2.39 (d, 2H, *J* = 8.0 Hz, H-9), 1.80 (m,1H, H-9), 1.62 (d, 3H, *J* = 8.0 Hz, H-8), 0.83 (d, 6H, *J* = 8.0 Hz, H-11 and H-12). ^13^C NMR (101 MHz, CDCl_3_) δ 155.16, 141.33, 137.11, 129.94, 126.87, 45.03, 37.60, 30.67, 30.15, 22.22, 20.20. HRMS (ESI+): m/z calculated for [(C_15_H_21_N_3_S)+H]^+^: 275.1456; found: 275.1454. Element analysis calculated: C, 65.42; H, 7.69; N, 15.26; S, 11.64; observed: C, 65.41; H, 7.69; N, 15.27; S, 11.65%.

#### 3.2.4. Synthesis of N-substituted aryl/alkyl 2-chloroacetamides **5a–5f**

Compounds **5a–5f** were prepared by the methods reported in [23]. Briefly, in an oven-dried round bottom flask, the *N*-substituted aryl/alkyl amines (12.0 moles) were dissolved in 10.0 mL of 5% Na_2_CO_3_ solution. Bromo acetyl bromide (12.0 mmoles) was introduced gradually into the above reaction mixture. The mixture containing round bottom flask was shaken gently till precipitation. Reaction progress as well as completion was continuously checked by TLC. After the completion, chilled water was added in flask. Precipitates of the product were filtered, dried and recrystallized using absolute ethanol.

#### 3.2.5. Synthesis of N-substituted 5-(1-(4-isobutylphenyl)ethyl)-1,2,4-triazole-2-yl- 2-sulfanyl acetamide derivatives **6a–6f**

Method A: Conventional method [24]. Different *N*-substituted 5-(1-(4-isobutylphenyl)ethyl)-1,2,4-triazole-2-yl-2-sulfanyl acetamide derivatives were synthesized in moderate yields by the stirring **4** (0.02 mol) with equimolar amount of different substituted aralkyl/alkyl/aryl 2-bromoacetamides **5a–f** using dichloromethane (15 mL) and NaH (0.01 mol) at room temperature for 10–30 h. Reaction progress as well as completion was continuously checked by TLC. After the completion, chilled water was added in flask to obtain precipitates of product. Precipitates of the product were filtered, dried and recrystallized using absolute ethanol.

Method B: Ultrasound-assisted method. Different *N*-substituted 5-(1-(4-isobutylphenyl)ethyl)-1,2,4-triazole-2-yl-2-sulfanyl acetamide derivatives were synthesized in good yields by the dissolving **4** (0.02 mol) in DCM (5 mL). NaH (0.01 mol) was added in the reaction mixture and it was stirred for 20 min. Equimolar amount of different substituted aralkyl/alkyl/aryl 2-bromoacetamides **5a–5f** was added and the reaction mixture was sonicated for 30–90 min at 45–55 °C (Figure 1). Reaction progress as well as completion was continuously checked by TLC. After the completion, chilled water was added in flask to obtain precipitates of product. Precipitates of the product were filtered, dried and recrystallized using absolute ethanol.

##### N-(4-Methylphenyl)-2-((5-(1-(4-isobutylphenyl)ethyl)-4-methyl-4H-1,2,4-triazol-3-yl)thio)acetamide (**6a**)

White amorphous solid, Yield 83%, m.p. 91–93 °C, IR: ν (cm^−1^): 3268.55, 1700.05, 1518.08, 1477.27, 1316.76, 1079.0, 758.83. ^1^H NMR (400 MHz, CDCl_3_) δ 10.78 (s, 1H, N-H), 7.55 (d, 2H, *J* = 8.0 Hz, H-2″, H-6″), 7.38 (d, 2H, *J* = 12.0 Hz, H-3 and H-5), 7.07 (d, 2H, *J* = 8.0 Hz, H-3″, H-5″), 7.03 (d, 2H, *J* = 8.0 Hz, H-2 and H-6), 4.09–4.04 (q, 1H, H-7), 4.02 (s, 2H, S-CH_2_), 3.20 (s, 3H, N-CH_3_), 2.66 (s, 3H, CH_3_ -4″), 2.42( d, 2H, *J* = 8.0 Hz, CH_2_-9), 1.83–1.74 (m, 1H, H-10), 1.77 (d, 3H, *J* = 8.0 Hz, CH_3_-8), 0.86 (d, *J* = 8.0 Hz, 6H, CH_3_-11, CH_3_-12). ^13^C NMR (126 MHz, DMSO) δ 166.12, 157.77, 148.86, 139.63, 139.63, 139.34, 135.15, 133.48, 132.37, 129.02, 128.14, 126.78, 125.37, 43.89, 37.23, 35.26, 30.50, 30.00, 21.87, 21.05, 17.69. HRMS (ESI+): m/z calculated for [(C_15_H_21_N_3_S)+H]^+^: 423.2119; found: 423.2211. Element analysis calculated: C, 68.21; H, 7.16; N, 13.26; S, 7.59; observed: C, 68.20; H, 7.17; N, 13.28; S, 7.57%.

##### N-(2,4-Dimethylphenyl)-2-((5-(1-(4-isobutylphenyl)ethyl)-4-methyl-4H-1,2,4-triazol-3-yl)thio)acetamide (**6b**)

Off white amorphous solid, Yield 89%, m.p. 94–96 °C. IR: ν (cm^−1^): 3282.85, 1669.22, 1522.77, 1433.67, 1366.60, 1078.08, 685.50. ^1^H NMR (500 MHz, DMSO) δ 9.55 (s, 1H, N-H), 7.24–7.23 (d, 1H, *J* = 5.0 Hz, H-6″), 7.09–7.05 (m, 4H, H-3, H-5 and H-2, H-6), 7.00 (s, 1H, H-3″), 6.95–6.93 (d, 2H, *J* = 10.0 Hz, H-5″), 4.38–4.29 (q, 1H, H-7), 4.01 (s, 2H, S-CH_2_), 3.26 (s, 3H, N-CH_3_), 2.41( d, 2H, *J* = 5.0 Hz, CH_2_-9), 2.24 (s, 3H, CH_3_-4″) 2.09 (s, 3H, CH_3_-2″) 1.84–1.76 (m, 1H, H-10), 1.60 (d, *J* = 5.0 Hz, 3H, CH_3_-8), 0.85 (d, *J* = 10.0 Hz, 6H, CH_3_-11, CH_3_-12). ^13^C NMR (126 MHz, DMSO) δ 165.83, 157.76, 148.85, 139.93, 139.36, 133.79, 132.90, 131.01, 129.02, 126.54, 124.26, 43.89, 37.24, 35.55, 29.99, 29.69, 20.47, 21.58, 20.80, 20.17, 17.37. HRMS (ESI+): m/z calculated for [(C_25_H_33_N_4_OS)+H]^+^: 437.2375; found: 437.2367. Element analysis calculated: C, 68.77; H, 7.39; N, 12.83; S, 7.34; observed: C, 68.75; H, 7.40; N, 12.81; S, 7.35%.

##### N-(2-Bromophenyl)-2-((5-(1-(4-isobutylphenyl)ethyl)-4-methyl-4H-1,2,4-triazol-3-yl)thio)acetamide (**6c**)

White amorphous solid, Yield 77%, m.p. 113–115 °C IR: ν (cm^−1^): 3345.65, 1688.84, 1516.33, 1464.88, 1320.42, 1085.06, 766.11. ^1^H NMR (500 MHz, DMSO) δ 9.90 (s, 1H, N-H), 7.75 (d, 1H, *J* = 10.0 Hz, H-6″), 7.50 (d, 1H, *J* = 10.0 Hz, H-3″), 7.33–7.31(t, 1H, *J* = 5.0 Hz and 10.0 Hz, H-5″), 7.21–7.18 (t, 1H, *J* = 5.0 Hz and 10.0 Hz, H-4″), 7.10–7.06 (m, 4H, H-2, H-3, H-5, H-6) 4.33–4.29 (q, 1H, H-7), 4.12 (s, 2H, S-CH_2_), 3.28 (s, 3H, N-CH_3_), 2.40 (d, 2H, *J* = 5.0 Hz, CH_2_-9), 1.79–1.76 (m, 1H, H-10), 1.60 (d, *J* = 5.0 Hz, 3H, CH_3_-8), 0.84 (d, *J* = 10.0 Hz, 6H, H-11, H-12). ^13^C NMR (126 MHz, DMSO) δ 166.15, 158.30, 148.85, 139.93, 139.60, 134.29, 129.27, 127.63, 126.24, 125.36, 43.89, 36.6, 35.56, 29.41, 29.12, 21.85, 20.77. HRMS (ESI+): m/z calculated for [(C_23_H_27_BrN_4_OS)+H]^+^: 487.1167; found: 487.1157. Element analysis calculated: C, 56.67; H, 5.58; N, 11.47; S, 6.56; observed: C, 56.66; H, 5.56; N, 11.48; S, 6.57%.

##### N-(2,3-Dimethylphenyl)-2-((5-(1-(4-isobutylphenyl)ethyl)-4-methyl-4H-1,2,4-triazol-3-yl)thio)acetamide (**6d**)

Off white amorphous solid, Yield 75%, m.p. 91–93°C. IR: ν (cm^−1^): 3235.73 (NH), 1635.22, 1514.79, 1444.90, 1389.08, 1075.04, 689.09. ^1^H NMR (500 MHz, DMSO) δ 9.68 (s, 1H, N-H), 7.10–7.00 (m, 7H, H-2, H-3, H-5, H-6, H-4″, H-5″ and H-6″), 4.32–4.30 (q, 1H, H-7), 4.02 (s, 2H, S-CH_2_), 3.26 (s, 3H, N-CH_3_), 2.39( d, 2H, *J* = 5.0 Hz, CH_2_-9), 2.23 (s, 3H, CH_3_-3″) 2.00 (s, 3H, CH_3_-2″) 1.80–1.76 (m, 1H, H-10) 1.60 (d, *J* = 5.0 Hz, 3H, CH_3_-8), 0.84 (d, *J*= 10.0 Hz, 6H, H-11, H-12). ^13^C NMR (126 MHz, DMSO) δ 165.86, 158.01, 148.85, 139.63, 139.35, 130.70, 129.03, 126.77, 125.36, 123.15, 43.90, 36.96, 35.55, 30.20, 29.99, 22.13, 20.47, 20.19, 13.77. HRMS (ESI+): m/z calculated for [(C_25_H_33_N_4_OS)+H]^+^: 437.2375; found: 437.2367. Element analysis calculated: C, 68.77; H, 7.39; N, 12.83; S, 7.34; observed: C, 68.75; H, 7.40; N, 12.81; S, 7.35%.

##### N-(4-Bromophenyl)-2-((5-(1-(4-isobutylphenyl)ethyl)-4-methyl-4H-1,2,4-triazol-3-yl)thio)acetamide (**6e**)

Off white amorphous solid, Yield 85%, m.p. 85–87°C. IR: ν (cm^−1^): 3233.65, 1688.25, 1516.33, 1465.83, 1318.23, 1076.24, 692.33. ^1^H NMR (500 MHz, DMSO) δ 10.40 (s, 1H, N-H), 7.51(d, 2H, *J* = 10.0 Hz, H-2″ and H-6″), 7.49 (d, 2H, *J* = 5.0 Hz, H-3″ and H-5″), 7.05 (d, 2H, *J* = 10.0 Hz, H-2 and H-6), 7.03 (d, 2H, *J* = 10.0 Hz, H-3 and H-5), 4.31–4.27 (q, 1H, H-7), 3.99 (s, 2H, S-CH_2_), 3.25 (s, 3H, N-CH_3_), 2.39 (d, 2H, *J* = 5.0 Hz, CH_2_-9), 1.81–1.74 (m, 1H, H-10), 1.58 (d, 3H, *J* = 12.0 Hz, CH_3_-8), 0.85 (d, *J* = 5.0 Hz, 6H, CH_3_-11 and CH_3_-12). ^13^C NMR (126 MHz, DMSO) δ 164.47, 158.01, 148.86, 139.34, 138.75, 138.24, 131.51, 129.0, 126.55, 120.89, 120.38, 43.91, 35.24, 31.58, 29.97, 22.44, 21.07. HRMS (ESI+): m/z calculated for [(C_23_H_27_BrN_4_OS)+H]^+^: 487.1167; found: 487.1157. Element analysis calculated: C, 56.67; H, 5.58; N, 11.47; S, 6.56; observed: C, 56.66; H, 5.56; N, 11.48; S, 6.57 %.

##### N-(3-Methylphenyl)-2-((5-(1-(4-isobutylphenyl)ethyl)-4-methyl-4H-1,2,4-triazol-3-yl)thio)acetamide (**6f**)

White amorphous solid, Yield 87%, m.p. 92–95°C. IR: ν (cm^−1^): 3266.73, 1694.15, 1522.99, 1477.87, 1312.86, 1077.08, 745.89. ^1^H NMR (500 MHz, DMSO) δ 9.68 (s, 1H, N-H), 7.43 (d, *J* = 15.0 Hz, 1H- H-6″), 7.39 (s, 1H, H-2″), 7.34 (dd, *J* = 10.0 Hz, 1H, H-5″), 7.08–7.05 (m, 5H, H-3, H-2, H-5, H-6 and H-4″), 4.33–4.28 (q, 1H, H-7), 4.04 (s, 2H, S-CH_2_), 3.27 (s, 3H, N-CH_3_), 2.40 (d, 2H, *J* = 10.0 Hz, CH_2_-9), 2.15 (s, 3H, CH_3_-3″), 1.83–1.76 (m, 1H, H-10), 1.60 (d, *J* = 10.0 Hz, 3H, CH_3_-8), 0.84 (d, *J* = 10.0 Hz, 6H, H-11, H-12). ^13^C NMR (126 MHz, DMSO) δ 166.12, 157.77, 148.86, 139.63, 139.34, 135.15, 133.48, 132.37, 129.02, 128.14, 126.78, 125.37, 43.89, 37.23, 35.26, 30.50, 30.00, 21.87, 21.05, 17.69. HRMS (ESI+): m/z calculated for [(C_15_H_21_N_3_S)+H]^+^: 423.2119; found: 423.2211. Element analysis calculated: C, 68.21; H, 7.16; N, 13.26; S, 7.59; observed: C, 68.20; H, 7.17; N, 13.28; S, 7.57%.

### 3.3. Experimental Procedures for Biological Activities

#### 3.3.1. Cell Culture and Treatment

Hepatic cancer cells were cultivated in Dulbecco’s modified eagle’s medium. Penicillin, 10% fetal bovine serum (FBS) and 100 μg/mL streptomycin were added to the above mixture at 37 °C. All media were maintained with 5% carbon dioxide-rich atmosphere.

#### 3.3.2. Determination of Cell Viability

The MTT assay was used to assess the viability of human liver cancer cells (HepG2 cells). In a 96 well plate, cancerous cells were cultured. For 48 h, liver cancer cells were treated with varying concentrations of drugs. After 48 h, MTT reagent (5 mg/mL) was added, and cells were incubated for 4 h at 37 °C. The formazan crystals were then dissolved in 150 L of Dimethyl sulfoxide (DMSO), and absorbance was measured in a microplate reader at 570 nm.

### 3.4. Molecular Docking

Molecular docking study was performed against five targets to understand the possible interaction mechanism of the molecules’ anticancer effect against HepG2 cell line. In particular, the results against Protein kinase B (Akt) and c-Kit Tyrosine Kinase (c-Kit) were excellent. For the docking studies, crystal structures of Protein Kinase B (Akt) and c-Kit Tyrosine Kinase (c-Kit) were obtained from the RCSB Protein Data Bank website (https://www.rcsb.org) under the PDB ID 2X39 and IT46, respectively [35]. The structures of the newly synthesized compounds were drawn using Chem Draw 20.1.1 and then exported to Marvin Sketch to create and minimize their 3D SDF structures. Before the docking process, the structure of the target proteins was checked, and errors in amino acid structures were corrected and optimized using Molegro Virtual Docker software [36]. The co-crystalized ligands at the crystal structure of proteins were chosen as the center of the grid box. They were re-docked again for the validation of the in silico process. New compounds were docked 10 times to the active sites of the target protein with Molegro Virtual Docker. The poses with the lowest binding energy and good interactions with the targets were chosen to analyze the interactions’ details. Discovery Studio Visualizer Software 2021 was utilized to visualize the 2D representations of the intermolecular interactions between the target and new derivatives.

## 4. Conclusions

A series of triazole-based acetamide derivatives **6a–f** were synthesized by utilizing ultrasound-assisted synthetic methodology and conventional protocols. All derivatives were obtained in low to moderate yields (60–75%) by using the conventional protocol in 16–26 h. By utilizing the ultrasound-assisted protocol, the compounds were obtained in good to excellent yields (75–89%) within a short period of time (40–80 min.) at 45–55 °C. After spectral characterization, all compounds were screened for their anti-proliferative potential against liver cancer cells and showed moderate to excellent anticancer activity, which varied according to the substitution on the aryl ring, as shown in Table 1. Among all derivatives, **6d** with a methyl substituent at the ortho- and meta-position of the aryl ring showed the best cell viability with an IC_50_ value of 13.004 μM. The mechanism of the inhibition of the compounds was further studied by in silico modelling. The results of molecular docking studies were in accordance with biological findings. The docked conformations of Kinase proteins revealed that **6d** has a superior docking score comparable with reference X39 and STI. We conclude that the ultrasound-assisted synthesis of triazole-based acetamides can be used to improve yields as well as the rate of reaction. Compound **6d** could be utilized as a lead compound in cancer therapy in the near future.

## Data Availability

Data are contained within the article.

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
