# Peer review of "Ultrasound Assisted Synthesis and In Silico Modelling of 1,2,4-Triazole Coupled Acetamide Derivatives of 2-(4-Isobutyl phenyl)propanoic acid as Potential Anticancer Agents"

_molecules, 2022, doi:10.3390/molecules27227984_

Round 1

Reviewer 1 Report

The authors described the ultrasound-assisted synthesis and in silico modelling of 1,2,4- triazole coupled acetamide derivatives of 2-(4-isobutyl phenyl)propanoic acid as potential anticancer agents. I consider that the manuscript meets all requirements to be published in “Molecules” after major revision. Additional suggestions and comments are included:   (1) See abstract. The authors mention that “using ultrasound radiations under different reaction conditions”. The optimization conditions should be specified. (2) See 2.1. Chemistry. h instead of hrs. It should be included in all manuscript. (3) See Scheme 1. It looks a little messy and confuse. Following several manuscripts, the reaction conditions should be included in the arrows.   (4) See Figure 1 and Table 1. It should be unified. For instance, the information in the Figure is not relevant. (5) See Figures 2 and 3. It should be moved to the Supplementary Material. It is not relevant in the manuscript. (6) See page 4, lines 110-121. The explanation of IR and NMR data should be improved. It is very important to include the range of most relevant signals taking into account all synthesized compounds.     (7) See Figure 6B. It is awful. The quality of the structure should be improved. (8) See Figure 8. The quality of the structure should be improved. (9) See 3. Materials and Method. HRMS, 1H, and 13C NMR data of compounds 2,3,4, and 5a-f should be included. NMR spectra should be included in the Supplementary Material. (10) See 3. Materials and Method for compounds 6a-j. 1H and 13C NMR reporting data should be carefully revised. For instance, (a) J in italic, (b) the unit of the coupling constant might be added, and (c) the J should be reported with one decimal after the point (i.e. J = 8.0 Hz instead of J = 8). The number of proton and carbons signals for each compound should be carefully verified. (11) See 3. Materials and Method for compounds 6a-j. According to the IUPAC nomenclature, the letters H and N should be in italic. It should be revised in all manuscript. (12) See 3.3.2. Determination of cell viability. It should be explained in this section. (13) See Figures S4, S7, S11, S14, S17, and S20. It seems an APT spectrum. It should be taken again due to the low resolution. (14) See Supplementary Material. HRMS spectra of compounds 6a-j should be included. (15) See Table 3. The standard should be specified.   (16) See 2.4. Molecular Docking. The authors should explain in a better form the selection of STAT3 for in-silico studies by using previous reports for triazole analogs.

Author Response

We are really grateful to the reviewer for his positive and encouraging comments. The manuscript has been substantially reformatted and modified based on suggestions provided by reviewer.

(1) See abstract. The authors mention that “using ultrasound radiations under different reaction
conditions”. The optimization conditions should be specified.
Response 1: The word “conditions” has been replaced with “Temperature” in revised manuscript according to reviewer’s recommendation.
(2) See 2.1. Chemistry. h instead of hrs. It should be included in all manuscript.
Response 2: We have revised our all manuscript according to reviewer’s recommendation and hrs has been replaced with h.
(3) See Scheme 1. It looks a little messy and confuse. Following several manuscripts, the reaction conditions should be included in the arrows.
Response 3: We have revised our Scheme 1 according to reviewer’s recommendation. All reaction conditions have been included within scheme in the arrows.

(4) See Figure 1 and Table 1. It should be unified. For instance, the information in the Figure is not relevant.
Response 4: Changes have been incorporated within revised manuscript according to reviewer’s recommendation.
(5) See Figures 2 and 3. It should be moved to the Supplementary Material. It is not relevant in the manuscript.
Response 5: We have removed Figures 2 and 3 according to reviewer’s recommendation.

(6) See page 4, lines 110-121. The explanation of IR and NMR data should be improved. It is very important to include the range of most relevant signals taking into account all synthesized compounds.
Response 6: We have revised our manuscript according to reviewer’s recommendation.
(7) See Figure 6B. It is awful. The quality of the structure should be improved.
Response 7: Dr. Adem from Turkey has revised our molecular docking studies and high quality structures have been included into manuscript according to reviewer’s recommendation.
(8) See Figure 8. The quality of the structure should be improved.
Response 8: Dr. Adem from Turkey has revised our molecular docking studies according to reviewer recommendations and high quality structures have been included into manuscript according to reviewer’s recommendation.
(9) See 3. Materials and Method. HRMS, 1H, and 13C NMR data of compounds 2,3,4, and 5a-f should be included. NMR spectra should be included in the Supplementary Material.
Response 9: Data has been added according to reviewer’s recommendation. NMR spectra of compound 2, 3 and 4 have been included in the Supplementary Material. Compound 5a-f were confirmed with TLC and they were further reacted with compound 4 to give 6a-f. Spectra of 6a-f have been included in the supplementary material.
(10) See 3. Materials and Method for compounds 6a-j. 1H and 13C NMR reporting data should be carefully revised. For instance, (a) J in italic, (b) the unit of the coupling constant might be added, and (c) the J should be reported with one decimal after the point (i.e. J = 8.0 Hz instead of J = 8). The number of proton and carbons signals for each compound should be carefully verified.
Response 10: We have carefully revised our manuscript according to reviewer’s
recommendation.
J is now in italic and reported with one decimal point after value and the unit of the coupling constant has been added. Proton numbers have been mentioned in Scheme 1, for further verification.
(11) See 3. Materials and Method for compounds 6a-j. According to the IUPAC nomenclature, the letters H and N should be in italic. It should be revised in all manuscript.
Response 11: We have carefully revised our manuscript according to reviewer’s recommendation. All letters H and N are in italic.
(12) See 3.3.2. Determination of cell viability. It should be explained in this section.
Response 12: Cell viability has been explained in revised manuscript.
(13) See Figures S4, S7, S11, S14, S17, and S20. It seems an APT spectrum. It should be taken again due to the low resolution.
Response 13: All figures have been taken again and included in the supplementary material according to reviewer’s recommendation.
(14) See Supplementary Material. HRMS spectra of compounds 6a-j should be included.
Response 14: HRMS spectra of compounds 6a-f have been included Supplementary Material (Page 16-21).
(15) See Table 3. The standard should be specified.
Response 15: Standard Sorafenib has been included in table. DMSO was used as negative control.
(16) See 2.4. Molecular Docking. The authors should explain in a better form the selection of STAT3 for in-silico studies by using previous reports for triazole.
Response 16: We have revised our manuscript according to reviewer’s recommendation.
We appreciate all your insightful comments. Thank you for taking the time and energy to help us improve the paper. The revised manuscript has been resubmitted to your journal. We are looking forward to hearing from you at your earliest convenience.
Sincerely Yours,
Dr. Samreen Gul Khan,
Assistant Professor/ Drug Design and Medicinal Chemistry Lab,
Department of Chemistry,
Government College University Faisalabad, Pakistan

Reviewer 2 Report

The study on triazole coupled acetamide derivaties sounds and display potential activity although the in vitro part supported by only one cell line (HepG2) and with in silico analysis. The results are promising, the authors may add one or teo additional cancer cell line to draw literature attention.

Author Response

We appreciate all your insightful comments. Thank you for taking the time and energy to help us improve the paper. We have plan to check these compounds against other cell lines like breast cancer and lung cancer. The revised manuscript has been resubmitted to your journal. We are looking forward to hearing from you at your earliest convenience.

Sincerely Yours,
Dr. Samreen Gul Khan,
Assistant Professor/ Drug Design and Medicinal Chemistry Lab,
Department of Chemistry,
Government College University Faisalabad, Pakistan.

Reviewer 3 Report

This article by Abourehab et. al. describes an environment friendly synthetic method to obtain molecules of biological importance containing triazole core as a continuation of the previously synthesized oxadiazole based polyvalent derivative. The ultrasound radiation assisted synthetic scheme affords series of N-substituted 1,2,4-triazole-2-thiol analogues in good overall yields via coupling reaction of 1,2,4-triazole of 2-(4-isobutylphenyl) propanoic acid  with respective electrophile. The synthesized compounds have been characterized using various spectroscopic methods and further investigated for their potential as anticancer effect against HepG2 cell lines and compared their activity with respect to their structural and electronic properties. The molecular design of the synthesized compounds has been on the basis of molecular hybridization and considering their potential as anticancer agents. The IC50 values, SAR studies, cell viability studies in conjugation with binding affinity studies using in silico modelling of all synthesized derivatives has been performed to support the results.

The paper is well written, and a suitable amount of data was presented by the authors to support the conclusions made. However, the authors should address the following points before the paper becomes publishable in the Journal-

1.     Though the introduction part is well written and focused on all the aspects pertaining to the importance and scope of the manuscript, I would suggest the authors change the order of introduction section to make it a continuous and easy to follow read. I would suggest that the authors consider starting with the second paragraph followed by the third then first then last two.

2.     The authors emphasize on the synthetic scheme being environment friendly, however, the reported method uses higher temperature (45 °C vs room temperature ) compared to the conventional synthetic method and the reagents/reaction media is the same. Though Ultrasound irradiation does shorten the reaction time very significantly and provides about 30% increase in reaction yield, the method should not be reported as being environment friendly just because the use of ultrasound irradiation.

3.     Can the authors comment on how the yield of the reaction is associated with the different aryl functionalities (in terms of EWG or EDG)?

4.     Line 132-134: The order of activity (2,3-Dimethyl phenyl ≥ 2,4-Dimethyl phenyl ≥ 3-Methyl phenyl ≥ 4-Methyl phenyl 133 ≥ 2-Bromo phenyl ≥ 4-Bromo phenyl) seems a bit confusing. As emphasized by the authors that derivatives 6d and 6b as having the highest potency as well as lowest of the IC50 values, shouldn’t the order be in the descending format using symbol  ‘>’ instead of ‘≥’.

5.     In the materials and methods section, the molecular mass of the compounds as obtained by HRMS is given only to the two decimals, please provide the HRMS mass values up to 4 decimals for all compounds (6a-6f).

6.     Supplementary information:

·      13C NMR spectra of compounds (6a-6f) are not phased correctly, please replace it with a properly phased NMR spectrum.

·      Please keep the supplementary information uniform, either put the chemical structure of all molecules over the 1H NMR spectrum with arrows indicating the NMR peaks associated with the chemical structure or remove from all.

Author Response

We are really grateful to the reviewer for his positive and encouraging comments. The manuscript has been substantially reformatted and modified based on suggestions provided by reviewer.
Comment 1. Though the introduction part is well written and focused on all the aspects pertaining to the importance and scope of the manuscript, I would suggest the authors change the order of introduction section to make it a continuous and easy to follow read. I would suggest that the authors consider starting with the second paragraph followed by the third then first then last two.
Response 1: Thank you for your valuable suggestions. All introduction part has been revised according to reviewer’s recommendation.
Comment 2. The authors emphasize on the synthetic scheme being environment friendly, however, the reported method uses higher temperature (45 °C vs room temperature) compared to the conventional synthetic method and the reagents/reaction media is the same. Though Ultrasound irradiation does shorten the reaction time very significantly and provides about 30%
increase in reaction yield, the method should not be reported as being environment friendly just because the use of ultrasound irradiation.
Response 2: We have replaced “ecofriendly with “economical” according to reviewer’s recommendation.
Comment 3. Can the authors comment on how the yield of the reaction is associated with the different aryl functionalities (in terms of EWG or EDG)?
Response 3: Electron donating groups improve stability of product and hence reaction complete early without any impurity with good yield of product. We have examined that compounds with electron donating moieties have fast rate of reaction as compared to compounds with electron withdrawing groups. All compounds were obtained in good yields.
Comment 4. Line 132-134: The order of activity (2,3-Dimethyl phenyl ≥ 2,4-Dimethyl phenyl ≥ 3-Methyl phenyl ≥ 4-Methyl phenyl 133 ≥ 2-Bromo phenyl ≥ 4-Bromo phenyl) seems a bit confusing. As emphasized by the authors that derivatives 6d and 6b as having the highest potency as well as lowest of the IC50 values, shouldn’t the order be in the descending format using symbol ‘>’ instead of ‘≥’.
Response 4: We have replaced “≥” with “>” according to reviewer’s recommendation. Data has been revised accordingly.
Comment 5. In the materials and methods section, the molecular mass of the compounds as obtained by HRMS is given only to the two decimals, please provide the HRMS mass values up to 4 decimals for all compounds (6a-6f).
Response 5: Dear reviewer HRMS mass values have been revised and data upto four decimal points have ben included. We have atached all Mass spectra in Supplementary Data (page 16-21).

Comment 6. Supplementary information: 13C NMR spectra of compounds (6a-6f) are not phased correctly, please replace it with a properly phased NMR spectrum.
Response 6a: 13C NMR spectra of all the compounds (6a-6f) are now phased
correctly and included in supplementary data.
· Please keep the supplementary information uniform, either put the chemical
structure of all molecules over the
1H NMR spectrum with arrows indicating the
NMR peaks associated with the chemical structure or remove from all.
Response 6b: Chemical structure of all molecules have been added over the 1H
NMR spectrum.
We appreciate all your insightful comments. Thank you for taking the time and energy to help us improve the paper. The revised manuscript has been resubmitted to your journal. We are looking forward to hearing from you at your earliest convenience.
Sincerely Yours,
Dr. Samreen Gul Khan,
Assistant Professor/Drug Design and Medicinal Chemistry Lab,
Department of Chemistry,
Government College University Faisalabad, Pakistan.

Round 2

Reviewer 1 Report

The authors described the ultrasound-assisted synthesis and in silico modelling of 1,2,4- triazole coupled acetamide derivatives of 2-(4-isobutyl phenyl)propanoic acid as potential anticancer agents. The authors performed most of the modifications suggested by reviewers. However, I consider that the manuscript meets all requirements to be published in “Molecules” after major revision. Additional suggestions and comments are included:   (1) See 3.2.1. Synthesis of methyl 2-(4-isobutylphenyl)propanoate (2). The NMR 1H reporting data is not correct. I did not see multiplicity, coupling constant, and integration of all signals.   (2) See 3.2.2. Synthesis of 2-(4-isobutylphenyl)propane hydrazide (3). The NMR 1H reporting data is not correct. I did not see multiplicity, coupling constant, and integration of all signals.   (3) 3.2.3. Synthesis of 5-(1-(4- isobutylphenyl) ethyl)-1,2,4-triazole -2-thiol (4). The NMR 1H reporting data is not correct. I did not see multiplicity, coupling constant, and integration of all signals.   (4) See 3. Materials and Methods. The calculated and found HRMS data should be specified with 4 decimals after the point and also error mass should be a maximum of 10 ppm. It should be carefully verified in all cases.    (5) See Supplementary Material. APT or 13C NMR spectra of compounds 2, 3, and 4 should be included in Supplementary Material. In addition, 13C NMR reporting data of compounds 2, 3, and 4 should be included in See 3. Materials and Methods.   (6) See 3. Materials and Methods. The elemental analysis calculated and found for compounds 2, 3, and 4 should be included. In addition, the data elemental analysis of compounds 6a-f should be verified because elemental analyzers can calculate only CHS or CHNS. However, I see data on halogen and oxygen ¿How do you determine it? The scientific rigor should be verified.     (7) See 3.1. General. All information should be included. For instance, I did not see the information on the elemental analyzer and high-resolution mass spectrometer.   
